# Imipramine Increases Norepinephrine and Serotonin in the Salivary Glands of Rats

**DOI:** 10.3390/biology13090679

**Published:** 2024-08-30

**Authors:** Kosuke Shirose, Masanobu Yoshikawa, Takugi Kan, Masaaki Miura, Mariko Watanabe, Mitsumasa Matsuda, Hiroyuki Kobayashi, Mitsuru Kawaguchi, Kenji Ito, Takeshi Suzuki

**Affiliations:** 1Department of Anesthesiology, School of Medicine, Tokai University, Isehara 259-1193, Japan; shirose.kosuke.k@tokai.ac.jp (K.S.); kantakugi@tokai.ac.jp (T.K.); 9j236559@tokai.ac.jp (M.W.); masui1327@star.tokai-u.jp (M.M.); ik3331@tokai.ac.jp (K.I.); takeshi-su@tokai.ac.jp (T.S.); 2Department of Clinical Pharmacology, School of Medicine, Tokai University, Isehara 259-1193, Japan; hkobayas@tokai.ac.jp; 3Tokyo Dental College, Tokyo 101-0061, Japan

**Keywords:** salivary gland, norepinephrine, serotonin, microdialysis, imipramine, monoamine reuptake inhibitor, xerostomia

## Abstract

**Simple Summary:**

In vivo microdialysis applied to rat submandibular glands showed that an increase in the norepinephrine and serotonin levels in the dialysate was primarily dependent on their release from the nerve endings. Either infusion of imipramine into interstitial fluids in rat submandibular glands through a dialysis probe or intraperitoneal administration of imipramine significantly and dose-dependently increased both norepinephrine and serotonin concentrations in the interstitial fluids.

**Abstract:**

Xerostomia induced by antidepressants such as imipramine has long been thought to be due to their anticholinergic effects. However, even antidepressants with low anticholinergic effects may have a high incidence of xerostomia. In salivary glands, norepinephrine activates alpha-adrenergic receptors in blood vessels and beta-adrenergic receptors in acinar cells, respectively, causing a decrease in the blood flow and an increase in the protein secretion, resulting in the secretion of viscous saliva with low water content and high protein content. A previous study demonstrated that perfusion of the submandibular glands of rats with serotonin significantly decreased saliva secretion. The results of the present study revealed the following: (1) that norepinephrine and serotonin, but not epinephrine nor dopamine, were detected in the interstitial fluids in rat submandibular glands; (2) that norepinephrine and serotonin concentrations in the dialysate was 4.3 ± 2.8 nM and 32.3 ± 19.6 nM at stable level, respectively; (3) that infusion with imipramine, a reuptake inhibitor of norepinephrine and serotonin, significantly and dose-dependently increased both norepinephrine and serotonin concentrations in the dialysate; and (4) that intraperitoneal administration of imipramine significantly increased both norepinephrine and serotonin concentrations in the dialysate. These results suggested that one of the mechanisms of xerostomia induced by reuptake inhibitors of norepinephrine and serotonin involves the activation of adrenergic and serotonin receptors in the salivary glands, respectively.

## 1. Introduction

As opposed to many organs, where the sympathetic and parasympathetic nerves work antagonistically, both nerves cooperate to enhance salivary gland function. Acetylcholine, released from postganglionic nerve endings in the salivary glands by parasympathetic excitation, stimulates the secretion of water and ions, i.e., serous saliva [1,2,3].

On the other hand, norepinephrine (NE), released from postganglionic sympathetic nerve endings within the salivary glands, has an effect on the salivary glands primarily stimulating protein secretion. Norepinephrine activates alpha-adrenergic receptors in blood vessels and beta-adrenergic receptors in acinar cells, respectively, causing a decrease in the blood flow and an increase in the protein secretion, resulting in the secretion of viscous saliva with low water content and high protein content [1,2,4].

Xerostomia or dry mouth is a collective term for diseases characterized by decreased salivary secretion. Saliva has various physiological effects, leading to caries and periodontitis in patients with xerostomia. In addition, xerostomia also contributes to decreased sensitivity to taste, difficulty in chewing and swallowing, and difficulty in speaking. Inflammation of the salivary glands and systemic diseases such as Sjogren’s syndrome are known as causes of xerostomia [5]. Xerostomia occurs even as a side effect of certain medical treatments, including irradiation for cancer treatment on the head and neck and drugs used to treat other diseases [4,5,6,7]. One particular problem with drug-induced xerostomia is that antidepressants are often administered over a long period of time, which leads to increased susceptibility to xerostomia [8]. Xerostomia induced by antidepressants has long been thought to be due to their anticholinergic effects. However, even antidepressants with low anticholinergic effects may have a high incidence of xerostomia [9].

Norepinephrine transporters (NET) and serotonin transporters (SERT) reuptake NE and serotonin (5-HT) in neuronal membranes as members of the Solute Carrier Family 6 (SLC6) that are widely distributed throughout the brain, respectively. Antidepressants such as imipramine inhibit NET and SERT in the brain and improve the function of these neurotransmitters, thereby exerting an antidepressant effect [10,11,12]. Adverse effects of inhibitors of NET and SERT are known to cause xerostomia.

Previous studies have shown that perfusion of the submandibular glands of rats with 5-HT decreased saliva and increased the protein content of saliva [13,14]. However, the effects of inhibitors of NET and SERT on the transporters of nerve endings in the salivary glands have not been elucidated.

The purpose of the present study is to evaluate changes in NE and 5-HT levels following injection of imipramine into the interstitial fluids of salivary glands of rats via dialysis probes or systemic administration of imipramine, an inhibitor of NET and SERT.

## 2. Materials and Methods

This study was conducted in accordance with the “Guidelines for the Care and Use of Animals for Scientific Purposes” of Tokai University. Approval for the protocol of this animal experiments was obtained from Tokai University (Approval No: 211051, 22204, 233018, 244005).

### 2.1. Animals

Male Wistar rats (from 8 to 9 weeks old, 230–280 g; CLEA Japan Inc., Tokyo, Japan) were reared under a 12 h light/dark cycle (light on: 7:00 a.m.) at a room temperature of 24–26 °C and a humidity 50–60%, with free access to food and water. Animals were acclimated to their new housing environment for at least one week.

### 2.2. Chemicals

The following were obtained from the sources indicated: imipramine hydrochloride (Tokyo Chemical Industry Co., Tokyo, Japan) and standard solution of monoamine for analysis (MA11-STD, Eicom, Kyoto, Japan). Unless otherwise indicated, all chemicals were purchased from the FUJIFILM Wako Chemical Co. (Osaka, Japan).

### 2.3. In Vivo Microdialysis in Anesthetized Rats

The interstitial fluids of the submandibular glands of rats were obtained by using in vivo microdialysis in the same manner as described previously [15]. Each sampling interval of 25 min (sample volume = 50 µL) was sufficient to collect enough volume for quantitative measurements of NE and 5-HT levels.

### 2.4. Quantification of Monoamines in Dialysates

The levels of monoamines in dialysates were measured using high performance liquid chromatography (HPLC) with electrochemical detection (HTEC-500, Eicom, Kyoto, Japan). The samples were separated on an octadecyl silyl (ODS) column (φ 3.0 mm × 150 mm; Eicompak SC-5ODS; Eicom, Kyoto, Japan). The quantification of NE and 5-HT was evaluated by using a peak area ratio relative to that of the following standards: norepinehrine hydrochloride (NE), serotonin hydrochloride (5-HT), 3-methoxy-4-hydroxyphenylglycol (MHPG), epinephrine hydrochloride (EPI), dihydroxyphenylacetic acid (DOPAC), normethanephrine hydrochloride (NM), dopamine hydrochloride (DA), 5-hydroxyindoleaceticacid (5-HIAA), isoprenaline hydrochloride (ISO), homovanillic acid (HVA), 3-methoxytyramine hydrochloride (3-MT).

### 2.5. Statistical Analyses

The results obtained are presented as means and standard deviations (SDs). Statistical analysis was conducted using GraphPad Prism 9.5.0 (GraphPad Software, San Diego, CA, USA). Nonparametric statistical comparisons were performed using a Mann–Whitney test or Friedman test followed by a Dunn’s multiple comparison test. A *p*-value of less than 0.05 was considered a statistically significant difference.

## 3. Results

### 3.1. Conditions for Optimal Dialysis Perfusion Rate In Vitro

Dialysates were obtained from five independent dialysis probes soaked in Ringer’s solution with 10 nM of NE and 5-HT at 37 °C at various perfusion rates (1 to 5 µL/min) (Figure 1). As the perfusion rate was increased from 1 µL/min to 5 µL/min, the relative recovery rate [(concentration in dialysate)/(concentration in test solution)] of NE and 5-HT decreased. The absolute recovery rate [(concentration in dialysate) × (perfusion rate)], on the other hand, showed a nonlinear increase, reaching an almost steady state at 2 µL/min. At 2 µL/min, the mean relative recoveries of NE and 5-HT were 54.8% ± 4.8% and 61.5% ± 2.9%, respectively, even when the NE and 5-HT concentrations in the test solution were varied (0.5–100 nM).

### 3.2. Monoamines in Dialysates from Submandibular Glands of Rats

Figure 2 shows chromatograms from a standard solution at 500 pg of each monoamine and 50 µL of dialysate samples. Norepinephrine and 5-HT, but not EPI nor DA, were detected in the dialysates. The basal concentrations of NE and 5-HT in dialysate samples from anesthetized rats, measured 125 min after probe implantation with a perfusion flow rate of 2 μL/min, were 4.3 ± 2.8 nM and 32.3 ± 19.6 nM, respectively (calculated from peak area, n = 24).

### 3.3. Time Course of Norepinephrine and Serotonin Levels in Dialysate after Probe Implantation

Figure 3 shows changes in NE and 5-HT levels in the dialysate of six rats collected at 25 min intervals over a period of 400 min. The levels of both NE and 5-HT were unstable and fluctuating during the first 100 min after probe implantation but decreased gradually to reach basal levels and remained stable up to 400 min.

### 3.4. Infusion of High-K^+^ Solution

To enhance monoamine release from nerve endings, Ringer’s solution containing 100 mM KCl (high-K^+^ solution) was infused in the interstitial fluids in the submandibular glands of five rats in each group through a microdialysis probe (Figure 4). Infusion with high-K^+^ solution significantly and transiently increased NE and 5-HT levels in the dialysate by 302 ± 102% and 167 ± 56%, respectively. The NE and 5-HT levels in the dialysate returned to the basal level at 100 min.

### 3.5. Infusion of Imipramine

Infusion of imipramine via dialysis probes significantly increased both the NE and 5-HT levels in the dialysates of six rats in each group in a dose-dependent manner (Figure 5). Imipramine had a median effective dose (ED_50_) of 1.86 × 10^−3^ M and 1.98 × 10^−3^ M in the value of area under the curve over 200 min (AUC_0–200min_) for the time course of the changes in NE and 5-HT levels, respectively (Figure 5C,D).

### 3.6. Intraperitoneal Administration of Imipramine

At 10 mg/kg of imipramine, which was considered to be clinically appropriate [16,17,18], significant increases in NE and 5-HT levels were observed in the dialysate of six rats (Figure 6).

## 4. Discussion

The present study demonstrated that imipramine significantly and dose-dependently increased the levels of both NE and 5-HT in the interstitial fluids in the submandibular glands of rats by either infusion in situ or intraperitoneal administration.

Excitation of sympathetic nerves releases NE from the endings, which causes contraction of the blood vessels via the alfa1-adrenergic receptor [1,2,19]. Protein secretion in salivary glands is regulated by cAMP production via beta-adrenergic receptors, resulting in activation of protein kinase A. In salivary glands, beta1-adrenergic receptors are highly expressed and respond well to NE released by the sympathetic nervous system. As a result, protein-rich viscous saliva with low water content is secreted. This is the reason why the sympathetic nervous system is excited by tense and stress, resulting xerostomia [4].

It is thought that 5-HT_1_, 5-HT_2_, 5-HT_3_, and 5-HT_4_ receptors are the primary 5-HT receptors involved in intestinal motility [20,21,22,23,24,25,26]. Of these, stimulation of 5-HT_2A_ receptors on intestinal smooth muscle enhances intestinal motility [27]. The other three 5-HT receptors are located on the cholinergic nerves in the intestinal tract. Stimulation of the 5-HT_1A_ receptor decreases acetylcholine release from the cholinergic nerves and inhibits intestinal motility, whereas stimulation of the 5-HT_3_ or 5-HT_4_ receptor increases acetylcholine release and promotes intestinal motility [28]. In the present study, a significant amount of 5-HT, approximately seven times more than NE at a steady state, was detected in the interstitial fluids in the submandibular glands of rats. The effects of 5-HT on the autonomic nervous system in the digestive tract suggest that potentiation of serotonergic effects affect autonomic activity of the salivary gland. Gene expressions of 5-HT_4B_ and 5-HT_7A_, which increase cAMP, as well as 5-HT_3_, an ionotropic receptor with permeability to calcium ions, has been shown in the salivary glands of rats [13,29]. Previous studies have shown that perfusion of the submandibular glands of rats with 5-HT through the external mandibular arteries activates 5-HT receptors, which increases cAMP production, resulting in a decrease in acetylcholine-induced salivation and an increase in the protein content of saliva [13,14]. 

Xerostomia induced by antidepressants has long been thought to be due to their anticholinergic effects. In general, tricyclic antidepressants have the strongest anticholinergic effects, followed by tetracyclic antidepressants with some anticholinergic effects, while selective serotonin reuptake inhibitors (SSRI), serotonin noradrenergic reuptake inhibitors (SNRI) and noradrenergic and specific serotonergic antidepressants (NaSSA) have little anticholinergic activity [8,9,30]. The stronger the anticholinergic effect, the more likely it has been thought to cause xerostomia. However, even antidepressants with low anticholinergic effects may have high incidence of xerostomia, in which case the mechanism of action is thought to be a decrease in autonomic activity of the salivary glands via the central nervous system [9]. The present study demonstrated that infusion of 10 mM of imipramine into interstitial fluids in the submandibular glands of rats and intraperitoneal administration of 10 mg/kg of imipramine to rats significantly increased te 5-HT contents approximately 22 and four times more than at steady state in the interstitial fluids, respectively. The results of the present study, together with those of previous studies, suggest that one of the mechanisms of xerostomia induced by inhibitors of NET and SERT involves activation of NE and 5-HT receptors in the salivary glands, respectively, in response to elevated levels of NE and 5-HT in their interstitial fluids. Further studies are needed to examine the relationship among saliva secretion, the pharmacokinetics of antidepressants in the salivary glands, and the amount of NE and 5-HT released into their intercellular fluids. In vivo microdialysis will be a powerful tool for these studies.

Assuming that the in vivo recovery rate is comparable to the in vitro recovery rate, the latter (about 55% and 62%) could be used to estimate the concentrations of NE and 5-HT in the interstitial fluids in the submandibular glands of rats. The estimated means of the concentration of NE and 5-HT recorded during the steady state were 7.8 and 52.1 nM, respectively, in the present study. Concentrations of 5-HT in plasma are kept low (<1 nM) by the metabolism of 5-HT to 5-HIAA by monoamine oxidase, as well as by the uptake of 5-HT into platelets via a SERT [31]. The estimated steady-state values of 5-HT concentration in dialysate were approximately 50 times greater than that of plasma. These results suggest that the levels of 5-HT in dialysate correspond to those derived from nerves in the salivary glands. Previous studies have found monoamines in homogenate obtained from the salivary glands of rats or mice [32,33]. The concentrations of NE, 5-HT, and DA in the homogenate from rats were approximately 1870, 410 and 24 nmol/g fresh weight, respectively. In the present study, no EPI nor DA was detected in the dialysate. The detection limit for monoamines, including EPI and DA according to the conditions of the microdialysis method applied in the present study, is approximately 10 pM, which suggests that the amount of EPI and DA in the intercellular spaces may have been below the detection limit.

Despite the rapid increase in NE and 5-HT levels after administration of antidepressants, the therapeutic effect is not observed until several weeks after dosing, indicating that the mechanism of action of antidepressants cannot be explained by the simple monoamine hypothesis and that more detailed mechanisms still need to be elucidated [34]. In fact, it has been reported that, after administration of desipramine, a tetracyclic antidepressant, or sibutramine, an SNRI, for 28 days, the amount of protein in the saliva decreased [35,36]. More detailed mechanistic insights remain to be elucidated in order to understand the possible antidepressant side effects on the salivary glands.

Whereas existing studies have analyzed changes in salivary flow and protein secretion due to the central and systemic effects of antidepressants [35,37,38], this study focused on quantitative changes in neurotransmitters in the salivary glands. Future studies are needed to clarify whether changes in these neurotransmitter levels in the interstitial fluids in the submandibular glands are associated with reduced salivary flow or changes in the biochemical composition of saliva. In addition, the effects of imipramine on NE and 5-HT levels in the submandibular glands, as well as the parotid and sublingual glands, will also be analyzed.

## 5. Conclusions

The present study demonstrated that imipramine, a monoamine reuptake inhibitor, increased NE and 5-HT levels in the interstitial fluids of the submandibular glands of rats, where the sympathetic nerve endings exist, by either its infusion or intraperitoneal administration.

## Figures and Tables

**Figure 1 biology-13-00679-f001:**
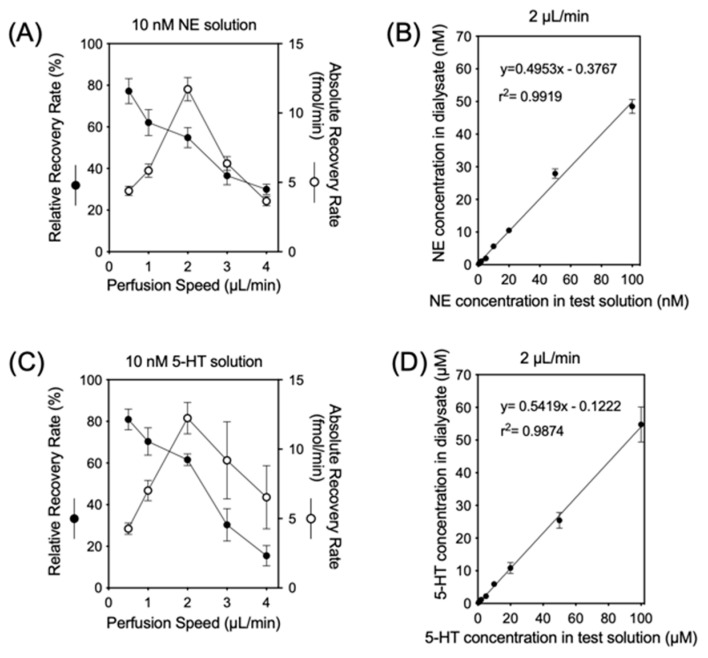
The upper left panel (**A**) and lower left panel (**C**) indicate the relationship between perfusion speed and relative (●) or absolute (○) recovery rates in vitro with 10 nM of NE or 5-HT in testing solution, respectively. The upper right panel (**B**) and lower right panel (**D**) indicate the relative recovery rate with different NE and 5-HT concentrations in testing solution using a 2 µL/min perfusion speed.

**Figure 2 biology-13-00679-f002:**
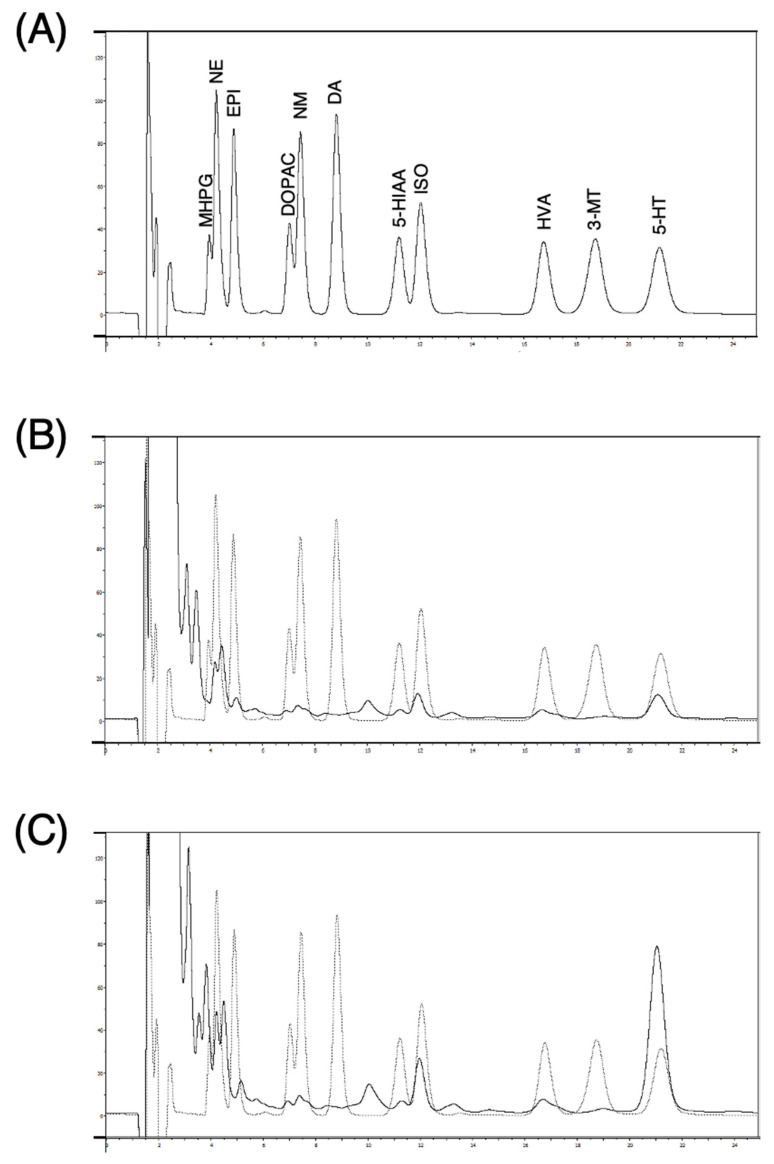
Representative chromatograms obtained from standard solution containing 500 pg monoamines each (**A**) and a microdialysis sample before (**B**) and after (**C**) infusion of 1 mM imipramine. A standard solution of monoamines for analysis contained 3-methoxy-4-hydroxyphenylglycol (MHPG), norepinehrine hydrochloride (NE), epinephrine hydrochloride (EPI), dihydroxyphenylacetic acid (DOPAC), normethanephrine hydrochloride (NM), dopamine hydrochloride (DA), 5-hydroxyindoleaceticacid (5-HIAA), isoprenaline hydrochloride (ISO), homovanillic acid (HVA), 3-methoxytyramine hydrochloride (3-MT), and serotonin hydrochloride (5-HT).

**Figure 3 biology-13-00679-f003:**
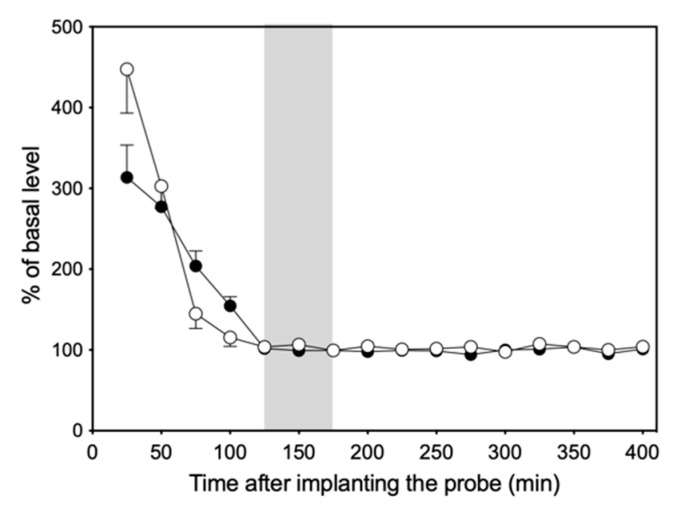
Change in NE (●) and 5-HT (○) levels in dialysate over time after probe implantation. The gray portion represents the three fractions at the basal level.

**Figure 4 biology-13-00679-f004:**
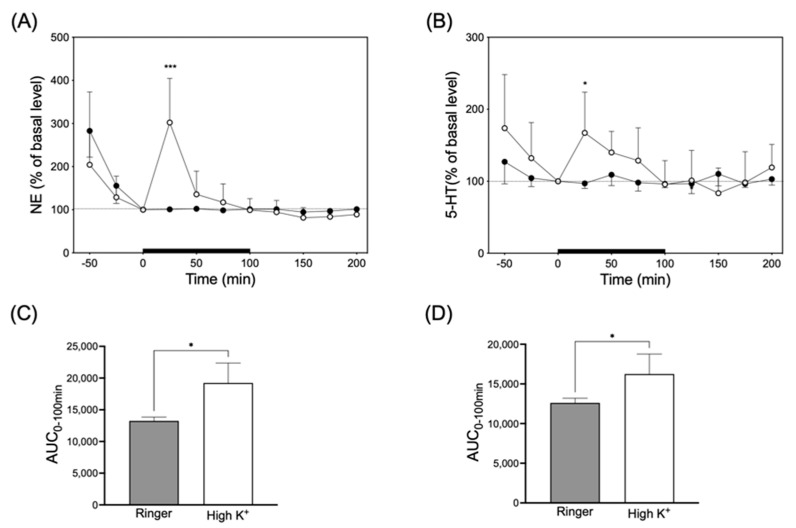
Left panel (**A**) and right panel (**B**) indicates NE and 5-HT release in response to infusion of high-K^+^ (○) or Ringer’s solution (●), respectively. Significantly different from basal level at each time according to Dunn’s post hoc test following Friedman test; * *p* < 0.05 and *** *p* < 0.001. Lower left (**C**) and right panel (**D**) indicate the AUC_0–100min_ for values of NE and 5-HT released in the upper panel (**A**) and (**B**), respectively. Significantly different from Ringer’s solution according to Mann–Whitney test; * *p* < 0.05.

**Figure 5 biology-13-00679-f005:**
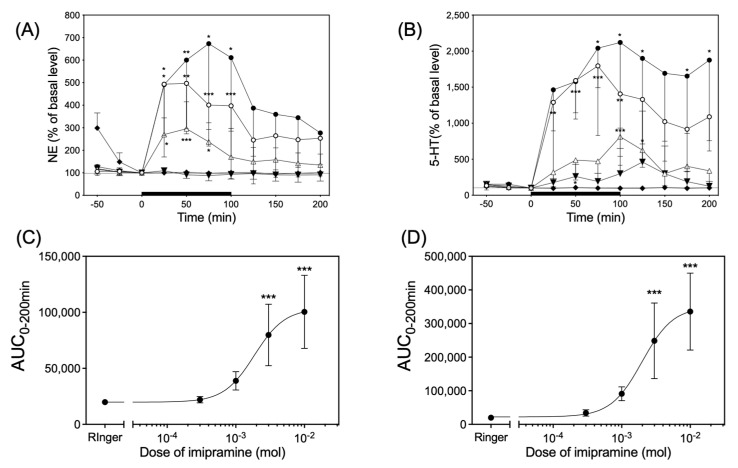
The upper left panel (**A**) and right panel (**B**) indicates time course of the NE and 5-HT release in response to infusion of 0.3 mM (▼), 1 mM (△), 3 mM (○), and 10 mM (●) of imipramine or Ringer’s solution (◆), respectively. The solid black bar indicates the length of application of imipramine solution through the dialysis probe. Significantly different from basal level at each time according to Dunn’s post hoc test following Friedman test; * *p* < 0.05, ** *p* < 0.01, and *** *p* < 0.001. The lower left (**C**) and right panel (**D**) indicate the AUC_0–200min_ for the values of NE and 5-HT released in upper panel (**A**) and (**B**), respectively. Significantly different from Ringer’s solution according to Dunn’s post hoc test following Friedman test; *** *p* < 0.001.

**Figure 6 biology-13-00679-f006:**
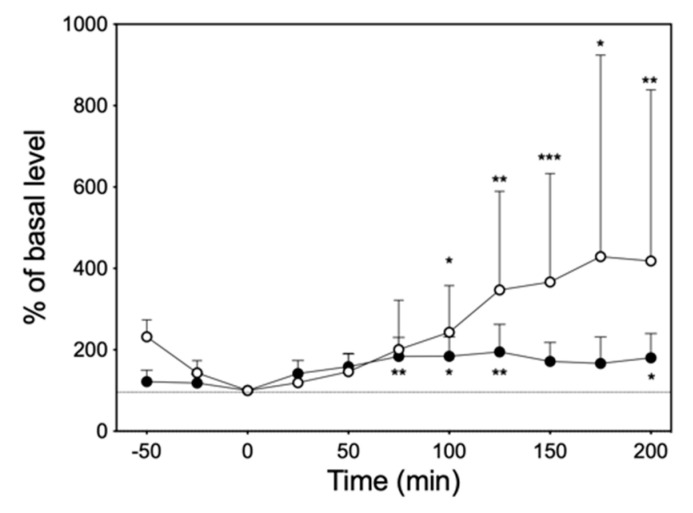
The time course of changes in NE (●) and 5-HT (○) levels in the dialysate after intraperitoneal administration of 10 mg/kg imipramine. The dashed line represents the 100% of basal level. Significantly different from basal level at each time according to Dunn’s post hoc test following Friedman test; * *p* < 0.05, ** *p* < 0.01, and *** *p* < 0.001.

## Data Availability

The data that support the findings of this study are available from the corresponding author, [M.Y.] upon reasonable request.

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
