# Peer review of "Imipramine Increases Norepinephrine and Serotonin in the Salivary Glands of Rats"

_biology, 2024, doi:10.3390/biology13090679_

Round 1

Reviewer 1 Report

Comments and Suggestions for Authors

I congratulate the authors on the idea, its implementation, and the well-written article. The proposed methodologies are well described and appropriate for the stated objectives. The results are correctly presented, and the discussion is interesting, completing the work as a whole. 

•The objective of the research is to investigate how imipramine administration affects norepinephrine and serotonin levels in the interstitial fluids of rat salivary glands. The study provides substantial evidence suggesting that xerostomia induced by imipramine may result from local mechanisms involving elevated concentrations of norepinephrine and serotonin. Nevertheless, the study does not clarify whether these changes in neurotransmitter levels are linked to a decrease in salivary flow or alterations in the biochemical composition of saliva. Given that xerostomia is a subjective sensation of dry mouth potentially attributable to either reduced salivary flow or changes in biochemical composition, further investigation is needed to elucidate the specific mechanisms.

• The topic is both unprecedented and highly relevant. It demonstrates that imipramine has a localized effect on the submandibular glands, as evidenced by increased levels of norepinephrine and serotonin, which can lead to glandular disorders such as hyposalivation or alterations in the salivary biochemical composition. However, a limitation of the study is that the parotid and sublingual glands were not investigated, which would have provided a more comprehensive understanding of imipramine's effects on the entire salivary gland system. Including these glands could have deepened the analysis and revealed whether the same effects observed in the submandibular glands also occur in other parts of the glandular system, offering a more complete view of the drug's implications.

While there has been extensive research on the systemic effects of imipramine, its direct impact on the salivary glands and neurotransmitter levels within these tissues remains unexplored. Understanding these interactions is crucial, as it may uncover previously unknown adverse effects that could influence salivary function.

• While most existing research focuses on the central and systemic effects of imipramine, this study highlights its local effects on the salivary glands, thereby broadening our understanding of the mechanisms through which imipramine can cause xerostomia.

• The authors should consider some specific improvements to the methodology. First, it is important to standardize all acronyms used throughout the study to ensure clarity and consistency. Additionally, the number of animals used was not specified, which is essential for assessing the study's statistical power. Another point to consider is the need for a clear justification of the 10 mg/kg dose of intraperitoneal imipramine; explaining this dosage choice is necessary to understand the relevance and applicability of the results. Finally, with regard to the statistical analysis, the authors should mention the normality test used, which is crucial for validating the appropriateness of the statistical methods applied to the data.

•The conclusions are consistent with the evidence and arguments presented and address the main question posed. The results are discussed coherently and logically, linking the observations made during the study with the initial hypothesis. The conclusions adequately reflect the findings and provide a clear answer to the central research question, demonstrating the relevance of imipramine's effects on neurotransmitter levels in the salivary glands and the potential consequences for glandular function.

•The references are appropriate. They encompass a wide range of relevant studies that support and contextualize the research, addressing both the pharmacology of antidepressants and the physiology of salivary glands. However, I suggest that the authors include a discussion of the paper 'Assessment of Redox State and Biochemical Parameters of Salivary Glands in Rats Treated with the Anti-Obesity Drug Sibutramine Hydrochloride,' which investigated the effects of sibutramine hydrochloride, a serotonin and norepinephrine reuptake inhibitor. This would provide better context for understanding the adverse effects that can occur in the salivary glands.

Author Response

  • I congratulate the authors on the idea, its implementation, and the well-written article. The proposed methodologies are well described and appropriate for the stated objectives. The results are correctly presented, and the discussion is interesting, completing the work as a whole.

Response: We thank the reviewer for his/her kind remark.

  • The objective of the research is to investigate how imipramine administration affects norepinephrine and serotonin levels in the interstitial fluids of rat salivary glands. The study provides substantial evidence suggesting that xerostomia induced by imipramine may result from local mechanisms involving elevated concentrations of norepinephrine and serotonin. Nevertheless, the study does not clarify whether these changes in neurotransmitter levels are linked to a decrease in salivary flow or alterations in the biochemical composition of saliva. Given that xerostomia is a subjective sensation of dry mouth potentially attributable to either reduced salivary flow or changes in biochemical composition, further investigation is needed to elucidate the specific mechanisms.

Response: We appreciate the reviewer’s suggestion and have now included such a sentence in the text. Please see page 9, lines 271-274. “Future studies are needed to clarify whether changes in these neurotransmitter levels in the interstitial fluids in the submandibular glands are associated with reduced salivary flow or changes in the biochemical composition of saliva.”

  • The topic is both unprecedented and highly relevant. It demonstrates that imipramine has a localized effect on the submandibular glands, as evidenced by increased levels of norepinephrine and serotonin, which can lead to glandular disorders such as hyposalivation or alterations in the salivary biochemical composition. However, a limitation of the study is that the parotid and sublingual glands were not investigated, which would have provided a more comprehensive understanding of imipramine's effects on the entire salivary gland system. Including these glands could have deepened the analysis and revealed whether the same effects observed in the submandibular glands also occur in other parts of the glandular system, offering a more complete view of the drug's implications.

Response: We appreciate the reviewer’s suggestion and have now included such a sentence in the text. Please see page 9, lines 274-275. “In addition, the effects of imipramine on NE and 5-HT levels in the submandibular as well as parotid and sublingual glands will also be analyzed.”

  • While there has been extensive research on the systemic effects of imipramine, its direct impact on the salivary glands and neurotransmitter levels within these tissues remains unexplored. Understanding these interactions is crucial, as it may uncover previously unknown adverse effects that could influence salivary function.

Response: We thank the reviewer for his/her kind remark.

  • While most existing research focuses on the central and systemic effects of imipramine, this study highlights its local effects on the salivary glands, thereby broadening our understanding of the mechanisms through which imipramine can cause xerostomia.

Response: We thank the reviewer for his/her kind remark.

  • The authors should consider some specific improvements to the methodology. First, it is important to standardize all acronyms used throughout the study to ensure clarity and consistency.

Response: We thank the reviewer for raising this important point. They should be written in capital letter. We have changed from Epi (Epinephrine), SSRIs (selective serotonin reuptake inhibitors), SNRIs (serotonin noradrenergic reuptake inhibitors) and NaSSAs (noradrenergic and specific serotonergic antidepressants) to EPI, SSRI, SNRI and NaSSA.

  • Additionally, the number of animals used was not specified, which is essential for assessing the study's statistical power.

Response: The reviewer raises an important matter in the original submission. We have corrected this statement in four places. Please see page 4, lines 133 and page 5, lines 140, 146, and 152.

  • Another point to consider is the need for a clear justification of the 10 mg/kg dose of intraperitoneal imipramine; explaining this dosage choice is necessary to understand the relevance and applicability of the results.

Response: We appreciate the reviewer’s suggestion and have now included such a sentence and some references in the text. Please see page 5, line 151-152. “At 10 mg/kg of imipramine, considered clinically appropriate [16-18], significant increases in NE and 5-HT levels were observed in the dialysate of six rats (Figure 6).”

  • Finally, with regard to the statistical analysis, the authors should mention the normality test used, which is crucial for validating the appropriateness of the statistical methods applied to the data.

Response: We thank the reviewer for raising this point, and correcting us. In the original manuscript we used two-way ANOVA in Figure 4, but changed to the Mann-Whitney test, a nonparametric test for AUC0-100min values for each of the two groups, for a more accurate statistical analysis even with the small number of animals used in each group.

  • The conclusions are consistent with the evidence and arguments presented and address the main question posed. The results are discussed coherently and logically, linking the observations made during the study with the initial hypothesis. The conclusions adequately reflect the findings and provide a clear answer to the central research question, demonstrating the relevance of imipramine's effects on neurotransmitter levels in the salivary glands and the potential consequences for glandular function.

Response: We thank the reviewer for his/her kind remark.

  • The references are appropriate. They encompass a wide range of relevant studies that support and contextualize the research, addressing both the pharmacology of antidepressants and the physiology of salivary glands. However, I suggest that the authors include a discussion of the paper 'Assessment of Redox State and Biochemical Parameters of Salivary Glands in Rats Treated with the Anti-Obesity Drug Sibutramine Hydrochloride,' which investigated the effects of sibutramine hydrochloride, a serotonin and norepinephrine reuptake inhibitor. This would provide better context for understanding the adverse effects that can occur in the salivary glands.

Response: We appreciate the reviewer’s suggestion and have now included such sentences and a reference [36] in the text. Please see page 9, line 265-268. “In fact, it has been reported that after administration of desipramine, a tetracyclic anti-depressant, or sibutramine, an SNRI, for 28 days, the amount of protein in the saliva decreased [35, 36]. More detailed mechanistic insights remain to be elucidated in order to understand the possible antidepressant side effects on the salivary glands.”

Reviewer 2 Report

Comments and Suggestions for Authors

Review of Biology-3147026

In this manuscript, the authors indicated that imipramine increases norepinephrine and serotonin in submandibular glands of rats by microdialysis method of salivary glands. This manuscript contains interesting and potentially useful results, however, a minor revision of manuscript is needed before it can be accepted for publication.

My comments are as follows.

The references need considerable revision. At present, they do not

conform to the Journal Guideline. For instance, reference 7 ‘…Oral Diseases 2003…’ should be ‘… Oral Dis 2003…’. There are other similar  problems that require attention. Wrong references numbers are 4, 5, 7, 9, 10, 11, 12, 13, 16, 26 and 30.

Author Response

  • In this manuscript, the authors indicated that imipramine increases norepinephrine and serotonin in submandibular glands of rats by microdialysis method of salivary glands. This manuscript contains interesting and potentially useful results, however, a minor revision of manuscript is needed before it can be accepted for publication.

Response: We thank the reviewer for his/her kind remark.

  • My comments are as follows.

The references need considerable revision. At present, they do not conform to the Journal Guideline. For instance, reference 7 ‘…Oral Diseases 2003…’ should be ‘… Oral Dis 2003…’. There are other similar  problems that require attention. Wrong references numbers are 4, 5, 7, 9, 10, 11, 12, 13, 16, 26 and 30.

Response: The reviewer raises an important matter in the original submission. We have corrected this statement. Please see references

Reviewer 3 Report

Comments and Suggestions for Authors

The authors provided the animal study discovering the potential mechanism of antidepressant-induced xerostomia. The topic of the manuscript is interesting and has a big clinical significance as the prevalence of drug-induced xerostomia remains a current problem in dentistry and general medicine.

The study is scientifically clear and the manuscript is interesting and well-written. However a few revisions may be recommended regarding Introduction and Discussion sections.

1.       Please, highlight the novelty of your results. Compare your research with the results of previously published articles investigating the antidepressants effect on salivary glands in rats. For instance:

Scarpace PJ, Koller MM, Rajakumar G. Desipramine desensitizes beta-adrenergic signal transduction in rat salivary glands. Neuropharmacology. 1992 Dec;31(12):1305-9. doi: 10.1016/0028-3908(92)90060-3. PMID: 1335132.

Henz SL, Cognato Gde P, Vuaden FC, Bogo MR, Bonan CD, Sarkis JJ. Influence of antidepressant drugs on Ecto-nucleotide pyrophosphatase/phosphodiesterases (E-NPPs) from salivary glands of rats. Arch Oral Biol. 2009 Aug;54(8):730-6. doi: 10.1016/j.archoralbio.2009.04.010. Epub 2009 May 26. PMID: 19473651.

Johnsson M, Winder M, Zawia H, Lödöen I, Tobin G, Götrick B. In vivo studies of effects of antidepressants on parotid salivary secretion in the rat. Arch Oral Biol. 2016 Jul;67:54-60. doi: 10.1016/j.archoralbio.2016.03.010. Epub 2016 Mar 19. PMID: 27023402.

Koller MM, Maeda N, Scarpace PJ, Humphreys-Beher MG. Desipramine changes salivary gland function, oral microbiota, and oral health in rats. Eur J Pharmacol. 2000 Nov 10;408(1):91-8. doi: 10.1016/s0014-2999(00)00770-6. PMID: 11070187.

2.       Paragraph 1 in the Introduction section: It would be better to place separate reference for each sentence, not 4 references at the end of the paragraph.

3.       In paraghraph explaining the regulation of salivation, if possible, change reference 1 for a more recent one.

Author Response

The authors provided the animal study discovering the potential mechanism of antidepressant-induced xerostomia. The topic of the manuscript is interesting and has a big clinical significance as the prevalence of drug-induced xerostomia remains a current problem in dentistry and general medicine.

The study is scientifically clear and the manuscript is interesting and well-written. However a few revisions may be recommended regarding Introduction and Discussion sections.

Response: We thank the reviewer for his/her kind remark.

  1. Please, highlight the novelty of your results. Compare your research with the results of previously published articles investigating the antidepressants effect on salivary glands in rats. For instance:

Scarpace PJ, Koller MM, Rajakumar G. Desipramine desensitizes beta-adrenergic signal transduction in rat salivary glands. Neuropharmacology. 1992 Dec;31(12):1305-9. doi: 10.1016/0028-3908(92)90060-3. PMID: 1335132.

Henz SL, Cognato Gde P, Vuaden FC, Bogo MR, Bonan CD, Sarkis JJ. Influence of antidepressant drugs on Ecto-nucleotide pyrophosphatase/phosphodiesterases (E-NPPs) from salivary glands of rats. Arch Oral Biol. 2009 Aug;54(8):730-6. doi: 10.1016/j.archoralbio.2009.04.010. Epub 2009 May 26. PMID: 19473651.

Johnsson M, Winder M, Zawia H, Lödöen I, Tobin G, Götrick B. In vivo studies of effects of antidepressants on parotid salivary secretion in the rat. Arch Oral Biol. 2016 Jul;67:54-60. doi: 10.1016/j.archoralbio.2016.03.010. Epub 2016 Mar 19. PMID: 27023402.

Koller MM, Maeda N, Scarpace PJ, Humphreys-Beher MG. Desipramine changes salivary gland function, oral microbiota, and oral health in rats. Eur J Pharmacol. 2000 Nov 10;408(1):91-8. doi: 10.1016/s0014-2999(00)00770-6. PMID: 11070187.

Response: We appreciate the reviewer’s suggestion and have now included such sentences in the text. Please see page 9, lines 261-275.

“Despite the rapid increase in NE and 5-HT levels after administration of antide-pressants, the therapeutic effect is not observed until several weeks after dosing, indi-cating that the mechanism of action of antidepressants cannot be explained by the simple monoamine hypothesis, and more detailed mechanisms are still needed to be elucidated [34]. In fact, it has been reported that after administration of desipramine, a tetracyclic antidepressant, or sibutramine, an SNRI, for 28 days, the amount of protein in the saliva decreased [35, 36]. More detailed mechanistic insights remain to be elucidated in order to understand the possible antidepressant side effects on the salivary glands.

Whereas existing studies have analyzed changes in salivary flow and protein secretion due to central and systemic effects of antidepressants [35, 37, 38], this study focused on quantitative changes in neurotransmitters in the salivary glands. Future studies are needed to clarify whether changes in these neurotransmitter levels in the interstitial fluids in the submandibular glands are associated with reduced salivary flow or changes in the biochemical composition of saliva. In addition, the effects of imipramine on NE and 5-HT levels in the submandibular as well as parotid and sublingual glands will also be analyzed.”

  1. Paragraph 1 in the Introduction section: It would be better to place separate reference for each sentence, not 4 references at the end of the paragraph.

Response: We appreciate the reviewer’s suggestion and have now changed the sentences and references in the text. Please see page 3, lines 41-50.

“   As opposed to many organs, where the sympathetic and parasympathetic nerves work antagonistically, both nerves cooperate to enhance salivary gland function. Acetylcholine, released from postganglionic nerve endings in the salivary glands by parasympathetic excitation, stimulates the secretion of water and ions, that is, serous saliva [1-3].

On the other hand, norepinephrine (NE), released from postganglionic sympathetic nerve endings within the salivary glands, has an effect on the salivary glands primarily stimulating protein secretion. Norepinephrine activates alpha-adrenergic receptors on blood vessels and beta-adrenergic receptors on acinar cells, respectively, causing a decrease in the blood flow and an increase in the protein secretion, resulting in the secretion of viscous saliva with low water content and high protein content [1, 2, 4].”

  1. In paraghraph explaining the regulation of salivation, if possible, change reference 1 for a more recent one.

Response: We appreciate the reviewer’s suggestion and have now changed the reference in the text. Please see page 3, lines 41-50.

“   As opposed to many organs, where the sympathetic and parasympathetic nerves work antagonistically, both nerves cooperate to enhance salivary gland function. Acetylcholine, released from postganglionic nerve endings in the salivary glands by parasympathetic excitation, stimulates the secretion of water and ions, that is, serous saliva [1-3].

On the other hand, norepinephrine (NE), released from postganglionic sympathetic nerve endings within the salivary glands, has an effect on the salivary glands primarily stimulating protein secretion. Norepinephrine activates alpha-adrenergic receptors on blood vessels and beta-adrenergic receptors on acinar cells, respectively, causing a decrease in the blood flow and an increase in the protein secretion, resulting in the secretion of viscous saliva with low water content and high protein content [1, 2, 4].”
